# Parasitoids of Chrysopidae Eggs in Sinaloa Mexico

**DOI:** 10.3390/insects11120849

**Published:** 2020-11-30

**Authors:** María de Lourdes Ramírez-Ahuja, Enrique Garza-González, Elijah J. Talamas, Mayra A. Gómez-Govea, Mario A. Rodríguez-Pérez, Patricia Zambrano-Robledo, Eduardo Rebollar-Tellez, Iram P. Rodríguez-Sanchez

**Affiliations:** 1Laboratorio de Fisiología Molecular y Estructural, Facultad de Ciencias Biológicas, Universidad Autónoma de Nuevo León, San Nicolás de Los Garza 66450, Mexico; lulu.ahuja@hotmail.com (M.d.L.R.-A.); mayragee@gmail.com (M.A.G.-G.); eddie_the_little@yahoo.com (E.R.-T.); 2Agrobionsa, Agrobiológicos del Noreste, Calle Rio Mocorito 575 pte, Guadalupe, Culiacán Rosales 80220, Mexico; egarzag9@hotmail.com; 3Florida State Collection of Arthropods, Division of Plant Industry, Florida Department of Agriculture and Consumer Services, Gainesville, FL 32608, USA; elijah.talamas@fdacs.gov; 4Centro de Biotecnología Genómica, Instituto Politécnico Nacional, Reynosa 88710, Mexico; marodriguez7862@hotmail.com; 5FIME-Centro de Investigación e Innovación en ingeniería Aeronáutica (CIIIA), Universidad Autónoma de Nuevo León, Av. Universidad s/n, Ciudad Universitaria, San Nicolás de los Garza 66455, Mexico; patricia.zambranor@uanl.edu.mx

**Keywords:** eggs parasitoids, *Telenomus*, crisopids, DNA barcoding

## Abstract

**Simple Summary:**

Four species of wasps, *Myartsevaia chrysopae, Telenomus lobatus*, *Telenomus tridentatus* and *Trichogramma atopovirilia,* are recorded as parasitoids of chrysopid eggs in Mexico, for the first time. The field survey was conducted in sorghum and corn in different locations in Sinaloa, Mexico. The identification of the parasitoids was determined by morphology, and for both *Telenomus* species the barcode region of the cytochrome oxidase 1 gene (CO1) was amplified and sequenced.

**Abstract:**

The eggs parasitoids *Myartsevaia chrysopae* (Crawford) (Hymenoptera: Encyrtidae), *Telenomus lobatus* Johnson, *Telenomus tridentatus* Johnson (Hymenoptera: Scelionidae) and *Trichogramma atopovirilia* Oatman and Platner (Hymenoptera: Trichogrammatidae) are reported for the first time or in new localities in Mexico. Their occurrence was first discovered in 2018 during a survey of parasitism on chrysopid eggs, conducted on *Sorghum bicolor* L. Moench (Poales: Poaceae) and *Zea mays* L. (Poales: Poaceae) in different locations in Sinaloa, Mexico. The identity of the parasitoids was determined by morphology and for both species of *Telenomus* the barcode region of the cytochrome oxidase 1 gene (CO1) was generated to facilitate molecular diagnosis of these species in future studies.

## 1. Introduction

Members of the insect family Chrysopidae (Neuroptera), commonly known as green lacewings, are important biological control agents as predators of several insect pests of agriculture. They have been reported preying on *Diaphorina citri* Kuwayama (Hemiptera: Liviidae) [1], whiteflies (Hemiptera: Aleyrodidae), *Parlatoria* (*Genaparlatoria*) *pseudaspidiotus* Lindinger (Hemiptera: Diaspididae) [2,3], and *Melanaphis sacchari* (Zehntner) (Hemiptera: Aphididae). In areas where *M. sacchari* occurs in Mexico, *Chrysoperla carnea* (Stephens) (Neuroptera: Chrysopidae) are released to control the pest [4]. Although *Ch. carnea* is the only species that is released against *M. sacchari*, at least eight species of Chrysopidae have been reported preying on *M. sacchari* under field conditions in Sinaloa, Mexico [5].

Many species of parasitoid wasps in the families Scelionidae, Encyrtidae and Trichogrammatidae are important natural enemies of agricultural pests, and some species are known to parasitize chrysopid eggs [6]. Only one species in the family Encyrtidae, *Myartsevaia chrysopae* (Crawford, 1913) (= *Ooencyrtus mexicanus* Myartseva and Shuvakhina), has been reported as an egg parasitoid of Chrysopidae in Mexico (Tamaulipas) [6,7]. Within Scelionidae, at least six species of *Telenomus* have been reported as eggs parasitoids of Neuroptera: *Te. chrysopae* Ashmead, *Te. ampullaceus* Johnson and Bin, *Te. suvae* Johnson and Bin, *Te. lobatus* Johnson and Bin, *Te. tridentatus* Johnson and Bin [8], and *Te. chrysoperlae* Loiácono [9]. Within Trichogrammatidae *Trichogramma tajimaense* Yashiro, Hirose and Honda and *Tr. semblidis* (Aurivillius) have been recorded as egg parasitoids of Neuroptera [10]. There have been few studies on the life histories of these parasitoids and their potential impact on the effectiveness of chrysopid as biological control agents [11,12] despite reports of more than 50% parasitism of *Chrysoperla* eggs by *Te. lobatus* [11]. To date, there is also almost no information about egg parasitoids of Chrysopidae in Mexico. Therefore, this study was conducted to identify parasitoids of chrysopid eggs, provide additional information about their biology, and determine their parasitism rates in sorghum and corn crops in Culiacan, Mexico.

## 2. Materials and Methods

### 2.1. Biological Samples

Parasitoids were obtained from field-collected chrysopid eggs on the leaves of *Sorghum bicolor* (L.) Moench (Poales: Poaceae) and *Zea mays* L. (Poales: Poaceae) in different locations in Sinaloa, Mexico during June and August 2018 (Table 1) and May and June 2019 (Table 2). Fifty leaves with eggs of Chrysopidae on their surfaces were cut and taken to the laboratory (RH 70%, 25 ± 2 °C). Each egg was individually placed into a 3 cm plastic box. Eggs of *Sitotroga cerealella* (Olivier) (Lepidoptera: Gelechiidae) were also placed in the box to feed the chrysopid larvae when they emerged. Eggs were kept under observation until the emergence of lacewings or adult parasitoids. Parasitoids that emerged were placed in 100% ethanol for morphological and molecular analyses. The number of eggs, emerged larvae, and parasitism rate were recorded.

### 2.2. Morphological Identification

We used the key of Johnson and Bin [8] to identify species of *Telenomus*, the keys of Pinto [13] for *Trichogramma*, and the keys of Myartseva and Shuvakhina [7] for Encyrtidae identification. For all of these groups, examination of the male genitalia is necessary for species-level determination. To prepare the slides of the male genitalia we followed the protocol of Polaszek and Kimani [14], which basically consists of permanent preparations in Canada balsam. The location of the emergence hole in the host egg and the color of the host egg are diagnostic for each of the species in this study. Once the males were associated with characteristics of the host egg, these were used to confirm the identity of the corresponding females. Larvae of Chysopidae were identified to species according to the keys of Tauber [15]. To identify chrysopid adults we used the key of Brooks [16] for *Chrysoperla* species and the key of Tauber [17] for *Ceraeochrysa.* All insect specimens were deposited in the Colección de Insectos Benéficos Entomófagos, Facultad de Ciencias Biológicas-Universidad Autónoma de Nuevo León.

### 2.3. Molecular Data

Genomic DNA was nondestructively isolated from the whole specimen using the Qiagen DNeasy kit (Hilden, Germany) as described by Giantsis [18]. Polymerase chain reaction (PCR) was carried out to amplify the mt CO1 (Cytochrome Oxidase Subunit 1) barcode region using the LCO1490 and HCO2198 primers [19]. The PCR was performed in a 20 µL reaction volume: 2 µL of DNA, 2 µL of 10× Qiagen PCR buffer containing 15 mM MgCl_2_, 0.9 µL of each primer, 0.6 µL of dNTPs (25 mM each), and 0.2 µL of (5 U/µL) Taq DNA Polymerase (Qiagen, Hilden, Germany), and 13.4 µL of H_2_O. The PCR conditions were as follows: 94 °C for 3 min, followed by 40 cycles of 94 °C for 30 s, 52 °C for 1 min, 72 °C for 1 min with a final extension at 72 °C for 10 min. All PCR products were electrophoresed through agarose gel (1%) and Sanger sequenced in both directions at the Florida Department of Agriculture Division of Plant Industry. Sequences were assembled and edited using Bioedit version 7. Voucher specimens were deposited at “Colección de Insectos Benéficos Entomófagos” (FCB-UANL) and the Florida State Collection of Arthropods (Gainesville-Florida). CO1 barcode sequences generated during this study were deposited in GenBank (Table 3).

## 3. Results

A total of 2311 chrysopid eggs were collected across all sites surveyed in Sinaloa during 2018–2019 (Table 1 and Table 2). Four species of parasitoids were recovered from 408 parasitized eggs: *Myartsevaia chrysopae*, *Telenomus lobatus*, *Telenomus tridentatus* and *Trichogramma atopovirilia*.

### 3.1. Morphological Identification

#### 3.1.1. *Myartsevaia chrysopae* (Crawford 1913)

Comments: *Myartsevaia chrysopae* can be recognized by the short digiti illustrated in Figure 1B. The host egg, viewed externally, appears as is seen in Figure 1A after the emergence of *M. chrysopae* (Female Figure 1C), which exits through the sides of the eggs but never through the apex as does *Te. lobatus* (Figure 2B).

Material examined: Mexico, Sinaloa, Culiacancito. 25-V-2019. Enrique Garza. 24 84 80 12–10754137. *Zea mays*. Directa de huevos de Crisopas. (CIBE 19-014) (7♀ 4♂); Mexico, Sinaloa, Ingenio El Dorado. 27-V-2019. M. L. Ramírez-Ahuja y Enrique Garza. 24.301390—107.377502. *Zea mays*. Directa de huevos de Chrysopidae (CIBE 19-015) (8♀ 4♂); Mexico, Sinaloa, Culiacán, Carretera Campo experimental Impa El Dorado. 29-V-2019. M. L. Ramírez-Ahuja y Enrique Garza. 24.367777–107.370003. *Zea mays*. Directa de huevos de Chrysopidae (CIBE 19-018) (2♀); México, Culiacán, Ingenio El Dorado. 29-V-2019. M. L. Ramírez-Ahuja y Enrique Garza. 24.342501–107.368332. *Zea mays*. Directa de huevos de Chrysopidae (CIBE 19-019) (2♀ 1♂); Mexico, Sinaloa, Paralelo. 38. 30-V-2019. M. L. Ramírez-Ahuja y Enrique Garza. 24.599167–107.481941. Sorgo para grano. Directa de huevos de Crisopas (CIBE 19-022) (1♀).

#### 3.1.2. *Telenomus lobatus* Johnson and Bin 1982 

Comments: *Te. lobatus* can be recognized by the digiti of the male genitalia with two long, widely separated teeth; lamina volsellaris in the form of a pair of strong, apically diverging rods or struts; aedeagal lobe which is extremely long and narrow (Figure 2C). The host egg viewed externally appears pink when parasitized (Figure 2A) and after the emergence of *Te. lobatus* (Female Figure 2D) it appears as in Figure 2B. *Telenomus lobatus* emerges from the apex of the egg (Figure 2B).

Material examined: Mexico, Sinaloa, El Dorado, Ingenio El Dorado. 19-VIII-2018. Enrique Garza González. 24.3012560 107 3772660. Sorgo forrajero. En huevos de Chrysopidae (CIBE 18-020) (3♀ 2♂); Mexico, Carretera Culiacán-El Dorado. 25-VIII-2018. Enrique Garza González y María de Lourdes Ramírez Ahuja. 24.505119 107.439178. Sorgo forrajero. En huevos de Chrysopidae (CIBE 18-023) (6♀ 1♂); Mexico, Culiacancito. 28-VIII-2018. Enrique Garza González y María de Lourdes Ramírez Ahuja. En maíz. Huevos de Chrysopidae (CIBE 18-027) (1♀ 2♂); México, Culiacancito. 25-V-2019. Enrique Garza. 24 84 80 12–10754137. *Zea mays*. Directa de huevos de Crisopas. (CIBE 19-014) (15♀ 7♂); México, Sinaloa, Ingenio El Dorado. 27-V-2019. M. L. Ramírez-Ahuja y Enrique Garza. 24.301390–107.377502. *Zea mays*. Directa de huevos de Crisopas (CIBE 19-015) (4♀ 3♂); Mexico, Culiacancito. 27-V-2019. M. L. Ramírez-Ahuja y Enrique Garza. 24.342222–107.368057. *Zea mays*. Directa de huevos de Crisopas. (CIBE 19-016) (6♀ 2♂); Mexico, Sinaloa, IMPA Campo Experimental. 29-V-2019. M. L. Ramírez-Ahuja y Enrique Garza. 24.572779-107.454445. *Zea mays*. Directa de huevos de Crisopas (CIBE 19-017) (2♀); Mexico, Sinaloa, Carretera Campo Experimental Impa, El dorado. 29-V-2019. M. L. Ramírez-Ahuja y Enrique Garza. 24.367777-107.370003. *Zea mays*. Directa de huevos de Crisopas (CIBE 19-018) (4♀ 2♂); Mexico, Sinaloa, Ingenio El Dorado. 29-V-2019. M. L. Ramírez-Ahuja y Enrique Garza. 24.342501 - 107.368332. *Zea mays*. Directa de huevos de Crisopas (CIBE 19-019) (3♀ 2♂); Mexico, Sinaloa, Bebelamas, 30-V-2019. M. L. Ramírez-Ahuja y Enrique Garza. 24.637222–107.273613. Sorgo forrajero. Directa de huevos de Crisopas (CIBE 19-020) (1♂); Mexico, Culiacán. 30-V-2019. M. L. Ramírez Ahuja y Enrique Garza. 24.633055-107.276665 *Zea mays*. Directa de huevos de Crisopas (CIBE 19-021) 155 (6♀ 3♂); Mexico, Paralelo 38. 30-V-2019. M. L. Ramírez-Ahuja y Enrique Garza. 24.599167- 107.481941. Sorgo para grano. Directa de huevos de Crisopas (CIBE 19-022) (2♀). Primary molecular vouchers Mexico, Sinaloa, Culiacancito, 28-VIII-2018, Enrique Garza y ML Ramírez-Ahuja, FSCA 00091168 (1♀)–FSCA 00091169 (1♂).

#### 3.1.3. *Telenomus tridentatus* Johnson and Bin 1982

Comments: *Telenomus tridentatus* (Figure 3D) is very similar to *Te*. *lobatus* and can be recognized by the male genitalia (Figure 3C): Each digitus with 3 massive, curved teeth; lamina volsellaris in the form of a pair of strongly melanized, apically diverging rods or struts; aedeagal lobe narrow and elongated. The host egg viewed externally appears yellow when parasitized (Figure 3A), and after the emergence of *Te*. *tridentatus* it becomes orange (Figure 3B).

Material examined: Mexico, Sinaloa, Guasave. 01, 02-VI-2018, 06-VI-2018, 10, 18, 20-VI-2018, 20, 25-VI-2018. Enrique Garza González (71 specimens); Mexico, Sinaloa, Tabalá (Bebelamas) 3.5 Km Sur de Tabalá. 23-VIII-2018. Enrique Garza González y María de Lourdes Ramírez Ahuja. 24.411330 107.092851. Sorgo forrajero. Directa de huevos de Chrysopidae (CIBE 18-022) (1♀ 2♂); Mexico, Sinaloa, Culiacancito. 25-V-2019. Enrique Garza. 24 84 80 12–10754137. *Zea mays*. Directa de huevos de Crisopas (CIBE 19-014) (34♀ 21♂); Mexico, Sinaloa, Ingenio El Dorado. 27-V-2019. M. L. Ramírez-Ahuja y Enrique Garza. 24.301390 -107.377502. *Zea mays*. Directa de huevos de Crisopas (CIBE 19-173015) (5♀ 5♂); Mexico, Culiacancito. 27-V-2019. M. L. Ramírez-Ahuja y Enrique Garza. 24.342222-107.368057. *Zea mays*. Directa de huevos de Crisopas (CIBE 19-016) (5♀ 3♂); Mexico, Sinaloa, IMPA Campo Experimental. 29-V-2019. M. L. Ramírez-Ahuja y Enrique Garza. 24.572779-107.454445. *Zea mays*. Directa de huevos de Crisopas (CIBE 19-017) (5♀ 6♂); Mexico, Sinaloa, Carretera Campo Experimental Impa, El dorado. 29-V-2019. M. L. Ramírez-Ahuja y Enrique Garza. 24.367777-107.370003. *Zea mays*. Directa de huevos de Crisopas (CIBE 19-018) (19♀ 7♂); Mexico, Sinaloa, Ingenio El Dorado. 29-V-2019. M. L. Ramírez-Ahuja y Enrique Garza. 24.342501-107.368332. *Zea mays*. Directa de huevos de Crisopas (CIBE 19-019) (2♀); Mexico, Sinaloa, Bebelamas, 30-V-2019. M. L. Ramírez- Ahuja y Enrique Garza. 24.637222-107.273613. Sorgo forrajero. Directa de huevos de Crisopas (CIBE 19-020) (1♀ 1♂); Mexico, Culiacán. 30-V-2019. M. L. Ramírez-Ahuja y Enrique Garza. 24.633055-107.276665. *Zea mays*. Directa de huevos de Crisopas (CIBE 19-021) (13♀ 13♂); Mexico, Sinaloa, Paralelo 38. 30-V-2019. M. L. Ramírez-Ahuja y Enrique Garza. 24.599167-107.481941. Sorgo para grano. Directa de huevos de Crisopas (CIBE 19-022) (5♀ 2♂). Primary molecular vouchers Mexico, Sinaloa, Culiacán, VII-2018, Enrique Garza, FSCA 00091170 (1♂).

#### 3.1.4. *Trichogramma atopovirilia* Oatman y Platner 1983 

Comments: Genital capsule broad, ratio of the width to length greater than 0.5; dorsal plate short and generally triangular with subangular apex; the volsellae becomes abruptly thinner in the apical half and with elongated apical spine ending in a point (Figure 4C) [13]. Host egg viewed externally appears like in (Figure 4A); *Trichogramma atopovirilia* (female Figure 4D) emerges through the sides of the eggs (Figure 4B) but never through the apex as does *Te*. *lobatus*.

Material examined: Mexico, Sinaloa, Guasave. Enrique Garza. 01,10, 18, 20, 25-VI-2018. Enrique Garza González (32 especimens); Mexico, Carretera Culiacán-El Dorado. 25-VIII-2018. Enrique Garza González y M. L. Ramírez Ahuja. 24.505119-107.439178. Sorgo forrajero. En huevos de Chrysopidae (CIBE 18-023) (6 specimens); Mexico, Culiacancito. 25-V-2019. Enrique Garza. 24 84 80 12-10754137. *Zea mays*. Directa de huevos de Crisopas. (CIBE 19-014) (5♀ 3♂); Mexico, Sinaloa, Ingenio El Dorado. 27-V-2019. M. L. Ramírez-Ahuja y Enrique Garza. 24.301390-107.377502. *Zea mays*. Directa de huevos de Crisopas (CIBE 19-015) (5♀ 4♂); Mexico, Carretera Campo Experimental Impa, El dorado. 29-V-2019. M. L. Ramírez-Ahuja y Enrique Garza. 24.367777-107.370003 *Zea mays*. Directa de huevos de Crisopas. (CIBE 19-018) (12♀ 4♂); Mexico, Sinaloa, Ingenio El Dorado. 29-V-2019. M. L. Ramírez- Ahuja y Enrique Garza. 24.342501-107.368332. *Zea mays*. Directa de huevos de Crisopas. (CIBE 19-019) (1♀); Mexico, Culiacán. 30-V-2019. M. L. Ramírez-Ahuja y Enrique Garza. 24.633055-107.276665. *Zea mays*. Directa de huevos de Crisopas (CIBE 19-021) (5♀ 1♂); Mexico, Sinaloa, Paralelo 38. 30-V-2019. M. L. Ramírez-Ahuja y Enrique Garza. 24.599167-107.481941. Sorgo para grano. Directa de huevos de Crisopas (CIBE 19-022) (3♀ 2♂).

#### 3.1.5. Diversity of Chrysopidae

During our field collections, we found four species of *Chrysoperla*: *Ch*. *comanche*, *Ch*. *externa*, *Ch*. *carnea*, *Ch*. *rufilabris*; and one species of *Cereaochrysa*: *Ce. valida.*

## 4. Discussion

Here we studied the parasitoids of chrysopid eggs in Sinaloa, Mexico, a region where *Ch*. *carnea* has been released as the most important biological control agent of *M*. *sacchari*. *Chrysoperla carnea* is massively produced in Mexico, as a nationwide strategy for the control of *M*. *sacchari* [20], and egg parasitoids can interfere with this biological control effort. Additionally, in Mexico, 21 laboratories commercially produce *Ch. carnea*, *Ch.rufilabris*, and *Ch*. *comanche* against different pest species [21], further emphasizing the importance of green lacewings in Mexican biocontrol. *Trichogramma* species are often endoparasitoids of lepidopteran eggs and are used worldwide in biological control programs. In Mexico *Tr*. *atopovirilia*, *Tr*. *exiguum* and *Tr. pretiosum* are commercially produced and massively released for the control of *Eoreuma loftini* and *Diatraea* spp. (Lepidoptera: Pyralidae) in sugarcane [21]. Before this work *Tr. atopovirilia* had not been reported as parasitoid of chrysopid eggs. Although its parasitism rate is low (Table 2 and Table 3), it must be considered as part of the biological control ecosystem. *Telenomus lobatus* and *Te*. *tridentatus* have been recorded in the United States attacking eggs of Chrysopidae and *Lomamyia flavicornis* Walker (Neuroptera: Berothidae) [8]. It has been reported that *Te*. *lobatus* can parasitize *Chrysoperla rufilabris* and *Ch*. *carnea* [8] with rates as high as 65% [12], and even higher rates have been observed [11]. In this work we did not determine the chrysopid eggs parasitized by *Te*. *lobatus* to the level of species. However, it should be noted that *Ch*. *rufilabris* and *Ch*. *carnea* were found in the places that we sampled in Sinaloa. It was not possible to determined that the highest percentage of parasitism was 35.57% in Guasave during 2018. In that year we collected eggs parasitized by *Telenomus* spp. and *Tr*. *atopovirilia*, but no *M. chrysopae* were collected. The coloration of eggs parasitized by *Te*. *lobatus* and *Tr. atopovirilia* are very similar when they are parasitized, but they can be distinguished by the emergence hole of the adult wasp. In *Te*. *lobatus* the hole is always at the apex of the egg whereas *Tr*. *atopovirilia* can exit from any side of the egg but not the apex. *Telenomus lobatus* and *Te*. *tridentatus* can be distinguished by male genitalia and egg color during development and after adult emergence. It was observed that *Telenomus* species prefer to oviposit in newly laid eggs, similar to what Ruberson [12] reported. Based on our observations, in areas in which *Te*. *lobatus* and *Te*. *tridentatus* occur, augmentative releases of *Chrysoperla carnea* should be conducted with mature chrysopid eggs to avoid parasitism. According to the results obtained in this study, parasitoid activity on populations of Chrysopidae must be considered in biological control and integrated pest management (IPM) programs because of the effect on predator efficacy. There is a need for more research on these parasitoids to know exactly which species of Chrysopidae are parasitized. There are now methods by which one can sequence residual DNA of both host and parasitoid from the empty egg. The methods described in Gariepy et al. [22] were recently used by Lomeli-Flores et al. [23] to define species-level trophic interactions. Future studies may employ this method to match DNA from the host egg with the adults and determine them at the species level.

## 5. Conclusions

This study showed that *M. chrysopae*, *Te. lobatus*, *Te. tridentatus*, and *Tr. atopovirilia* are parasitoids of Chrysopidae eggs in Culiacan, Mexico. This finding has important implications for the development of biological control and IPM programs using Chrysopidae against different pest species.

## Figures and Tables

**Figure 1 insects-11-00849-f001:**
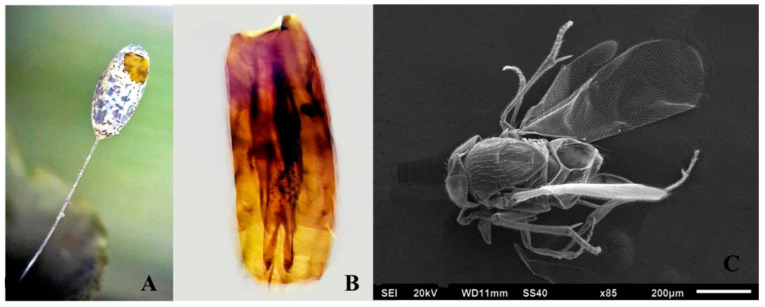
(**A**) Chorion of chrysopid egg following emergence of *M. chrysopae*, (**B**) Male genitalia, (**C**) Female *M. chrysopae.*

**Figure 2 insects-11-00849-f002:**
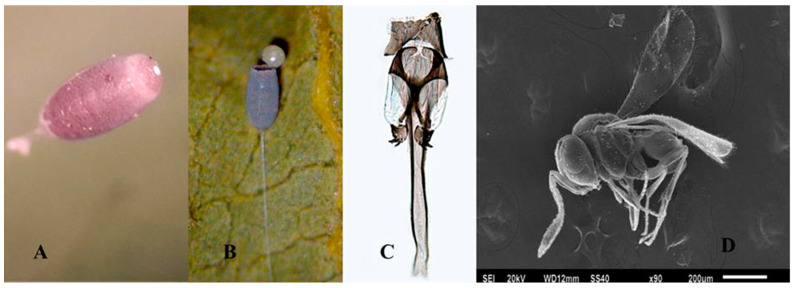
**(A**) Egg parasitized by *Telenomus lobatus*, (**B**) Chorion of Chrysopidae abandoned by *Te*. *lobatus*, (**C**) Male genitalia, (**D**) Female *Te*. *lobatus*.

**Figure 3 insects-11-00849-f003:**
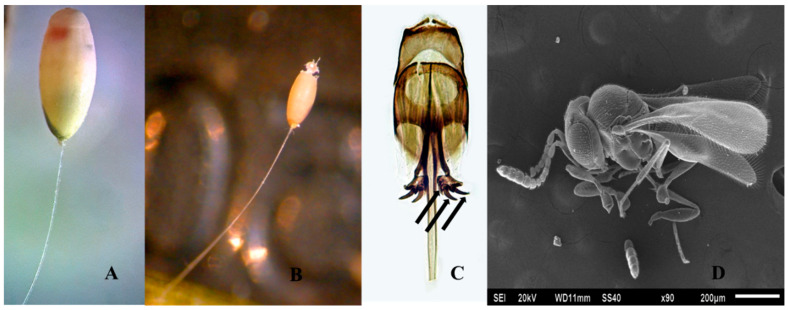
(**A**) Egg parasitized by *Te*. *tridentatus*, (**B**) Chorion of Chrysopidae abandoned by *Te*. *tridentatus*, (**C**) Male genitalia, medial tooth very close to aedeagal lobe and not clearly visible, (**D**) Female *Te*. *tridentatus*.

**Figure 4 insects-11-00849-f004:**
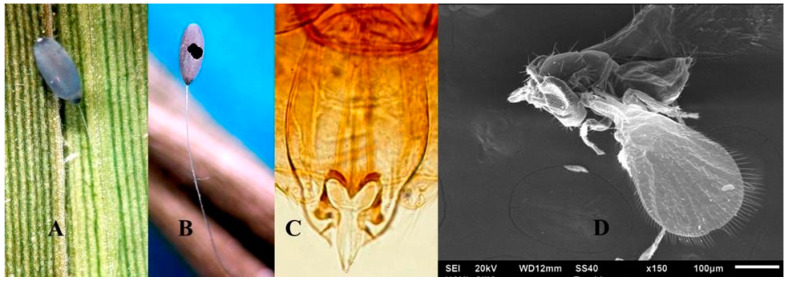
(**A**) Egg parasitized by *Trichogramma atopovirilia*, (**B**) Chorion of chrysopid egg parasitized by *Tr*. *atopovirilia*, (**C**) Male genitalia (photo by Fabián García), (**D**) Female of *Tr*. *atopovirilia.*

**Table 1 insects-11-00849-t001:** Collection dates, localities, and number of chrysopid eggs collected in field surveys in Sinaloa during 2018.

Locality	Coordinates	Date	Total Eggs	Parasitized Eggs	Parasitized by *Tr. atopovirilia*	Parasitized by *Telenomus*	Total Parasitism
Guasave	25.56745N 108.46756W	02-VI-2018	65	5	0	5 = 7.6%	7.6%
Guasave	25.56745N 108.46756W	06-VI-2018	44	5	0	5 = 11.3	11.3%
Guasave	25.56745N 108.46756W	10-VI-2018	65	19	8 = 12.3%	11 = 16.92%	29.23%
Guasave	25.56745N 108.46756W	18-VI-2018	54	13	3 = 5.5%	10 = 18.5%	24%
Guasave	25.56745N 108.46756W	20-VI-2018	42	7	0	7 = 16.7%	16.7%
Guasave	25.56745N 108.46756W	25-VI-2018	104	37	19 = 18.3%	18 = 17.3%	35.57%
Guasave	25.56745N 108.46756W	01-VI-2018	115	17	2 = 1.7%	15 = 13.4%	14.78%
El Dorado	24.3012560N 107.3772660W	19-VIII-2018	195	5	0	5 = 2.56%	2.56%
Tabalá	24.411330N 107.092851W	23-VIII-2018	60	3	0	3 = 5%	5%
El Dorado	24.354280N 107.339325W	25-VIII-2018	95	13	6 = 6.31%	7 = 7.36%	13.68%
Culiacancito	24.848012N 107.54137W	28-VIII-2018	27	3	0	3 = 11.11%	11.11%
Total			866	127	Overall parasitism rate: 14.66%		

**Table 2 insects-11-00849-t002:** Collection dates, localities, and number of chrysopid eggs collected in field surveys in Sinaloa during 2019.

Locality	Coordinates	Date	Total Eggs	Parasitized Eggs	Parasitized by *Tr. atopovirilia*	Parasitized by *Telenomus*	Total Parasitism
Culiacancito	24.848012N 107.54137W	25-V-2019	293	94	8 = 2.73%	75 = 25.59%	11 = 3.75%
Ingenio El Dorado	24.301390N 107.377502W	27-V-2019	143	38	9 = 6.29%	17 = 11.88%	12 = 8.39%
Culiacancito	24.342222N 107.368057W	27-V-2019	192	16	0%	16 = 8.33	0
IMPA Campo exp	24.572779N 107.454445W	29-V-2019	75	13	0%	13 = 17.33%	0
Carretera IMPA	24.367777N 107.370003W	29-V-2019	269	50	26 = 9.66%	22 = 8.17%	2 = 0.74%
Ingenio El Dorado	24.342501N 107.368332W	29-V-2019	71	11	2 = 2.81%	6 = 8.45%	3 = 4.22%
Bebelamas	24.637222N 107.273613W	30-V-2019	51	3	0	3 = 5.88%	0
Culiacán	24.633055N 107.276665W	30-V-2019	274	41	6 = 2.18%	35 = 12.77%	0
Paralelo 38	24.599167N 107.481941W	30-V-2019	77	15	5 = 6.49%	9 = 11.68	1 = 1.29%
Total			1445	281	Overall parasitism rate: 19.44%		

**Table 3 insects-11-00849-t003:** Accession number of CO1 sequences of *Telenomus* species.

Species	Collection Unit Identifier	GenBank Accession Number
*Telenomus lobatus*	FSCA 00091168	MT846080
*Telenomus lobatus*	FSCA 00091169	MT846081
*Telenomus tridentatus*	FSCA 00091170	MT846082

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
