# Peer review of "Parasitoids of Chrysopidae Eggs in Sinaloa Mexico"

_insects, 2020, doi:10.3390/insects11120849_

Round 1

Reviewer 1 Report

The manuscript is written in  poor English, particularly the Abstract. How many times the word parasitoids appears in the first sentence alone?

Title: why specify one locality (Culiacan) when it is said later that samples were from several localities? Replace with Sinaloa in the title.

Abstract: How can you say that Ooenyrtus mexicanus is for the first time reported from Mexico when it was originally described from Mexico and even named after the country?  Such blunders really show sloppiness in basic writing.

Besides, Ooencyrtus mexicanus is a current synonym of

Myartsevaia chrysopae (Crawford , 1913), see

https://www.nhm.ac.uk/our-science/data/chalcidoids/database/detail.dsml?FamilyCode=EE&ValFamTrib=&VALGENUS=Myartsevaia&VALSPECIES=chrysopae&VALAUTHOR=%28Crawford%29+&VALDATE=1913&ValidAuthBracket=true&TAXONCODE=Ooency+mexicM&HOMCODE=0&searchPageURL=indexValidName%2edsml%3fSpecies%3dmexicanus%26Familyqtype%3dequals%26Genusqtype%3dequals%26index%3dValidName%26Speciesqtype%3dequals&listPageURL=validName%2edsml%3fSpecies%3dmexicanus%26Genusqtype%3dequals%26Familyqtype%3dequals%26index%3dValidName%26Speciesqtype%3dequals

The authors mention that synonymy but yet use the name of the synonymized species instead!

Page 1: Diaphorina citri is (Hemiptera: Lividae), no in Homoptera: Psyllidae

Page 2: abbreviate Telemomus as T., not as Te.

Page 3.: Zea mays should be in Italics

References:  In several occasions, only one page is indicated when obviously there is a rane of those, for instance:

7. Myartseva, S. N.; Shuvakhina, E.Y. Species of the genus Ooencyrtus Ashmead (Hymenoptera, Encyrtidae), 285 lacewing egg parasites (Neuroptera, Chrysopidae) in North and South America. Entomologicheskoe 286 Obozrenie, 2004, 83: 253.

Polaszek is misspelled both in the citation and in the reference.

Overall, I find this contribution very interesting but in need of a very major revision, paying special attention to the following:

1) Very sloppy writing throughout, with a lot of inconsistencies and often in poor English (particularly the Abstract). Italics are not used when needed in some scientific names, all the Material examined sections need a much better and concise organization, (indicate the country first), primary molecular vouchers need to be marked in the Material examined.

2) Supporting DNA sequence data is also needed for the identification of the Trichogramma species. This is the best way to go now as it is easy to misidentify species of this genus morphologically.

3) The choice of scanning electron micrographs for the illustration of habitus of the parasitoids is very poor, it is best used for specific morphology details (having in mind that users will not have access to those so in reality the wasps would look a bit different). Instead, color habitus images should be taken using an Automontage or similar settings from well mounted, critical point dried or HMDS-ed specimens. From thise SEM images, one can only tell that these are encyrtids or scelionids, which is useless for identification or any purpose. Also, useful taxonomic details of their identifications to the species need to be thoroughly illustrated unless they are already available, and any differences between them and the previously described/redescribed specimens need to be indicated to, if there are some.

Reviewer 2 Report

I read with attention the submitted manuscript titled" Parasitoids od Chrysopidae eggs in Culiacan, Mexico".

In this paper the AA. consider four species of chrysopids newly records for Mexico and pursue the aim to provide morphological  data for recognition  and their characterization. However the shown results  are almost scanty to satisfy the aim because no any full morpho-systematic description of the adults of each recorded species has been reported , although the high number of the available specimens collected in the Mexican areas. Moreover, no identification of the four species of chrysopid hosts of the parasitoids  has been provided.

Here are the main suggestions, while others are shown in the attached file of mns:

 In the morho-systematic descriptions of parasitoids species (Results) I strongly  recommend to use the morphological characters terms that have to be conform to the requirements of the amended International Code of Zoological Nomenclature.

 In the paragraph  "Molecular Data"  (Methods)

  • please specify the type of sequencing performed; 
  • Did the AA. exclude the presence of pseudogenes (i.e. frame-shift mutations or stop-codons) through the translation in amino-acid of obtained sequences? I would suggest to check with MEGA software or other available tools;
  • In Results:
  • Why did not the AA. perform the molecular identification of all parasitoid species (i.e. Ooencyrtus mexicanus, Trichogramma atopovirilia)?
  • The sequences deposited are few in relation to the individuals emerged from the parasitized eggs;
  • Has any comparison been made with sequences available in database? 

 I suggest to improve the molecular part of the work by defining how the molecular data helped in the identification of species and the importance they have in the present work.

The chapters Discussion and Conclusions should be re-written and improved after the revision of data for Material and Methods and Results.

Also, please, check the whole references list , because several mistakes are present. The list could be improved.

Round 2

Reviewer 2 Report

See the notes in the revised new manuscript.
